# Learning Ordered Representations in Latent Space for Intrinsic Dimension Estimation via Principal Component Autoencoder

## Abstract

Autoencoders have long been considered a nonlinear extension of Principal Component Analysis (PCA). Prior studies have demonstrated that linear autoencoders (LAEs) can recover the ordered, axis-aligned principal components of PCA by incorporating non-uniform $\ell_2$ regularization or by adjusting the loss function. However, these approaches become insufficient in the nonlinear setting, as the remaining variance cannot be properly captured independently of the nonlinear mapping. In this work, we propose a novel autoencoder framework that integrates non-uniform variance regularization with an isometric constraint. This design serves as a natural generalization of PCA, enabling the model to preserve key advantages, such as ordered representations and variance retention, while remaining effective for nonlinear dimensionality reduction tasks.

## 1 Introduction

Principal Component Analysis (PCA) remains one of the most widely used techniques for dimensionality reduction due to its simplicity, interpretability, and ability to produce ordered, variance-preserving latent representations. As a natural extension, autoencoders have been proposed to generalize PCA into the nonlinear regime by leveraging neural networks to learn complex mappings. However, autoencoders suffer from a major limitation in practice: the bottleneck (latent) dimension must be chosen in advance, which requires prior knowledge or expensive tuning.

For linear autoencoders (LAEs), previous work has demonstrated that introducing non-uniform $\ell_2$ regularization (Bao et al., 2020) or modifying the loss function (Oftadeh et al., 2020) enables the model to recover the principal components of PCA in an axis-aligned and variance-ordered manner. However, extending these desirable properties to nonlinear autoencoders remains challenging. The key difficulty lies in the fact that variance allocation becomes entangled with the nonlinear transformation itself, making it difficult to preserve the global variance structure inherent to PCA.

In this work, we propose a nonlinear autoencoder framework that addresses this challenge by integrating non-uniform variance regularization with an isometric constraint. This combination allows the model to retain PCA-like properties, such as ordered representation and variance control, while benefiting from the flexibility of nonlinear mappings. We theoretically and empirically verify that the proposed method achieves effective nonlinear dimensionality reduction while preserving meaningful variance structure across various datasets. Compared to a conventional autoencoder, the main advantage of our model is that one only needs to assume a (sufficiently large) upper bound for the bottleneck dimension; then, like PCA, one can select the appropriate number of latent dimensions post hoc based on the variance captured by each learned component.

## 2 Related Work

The fundamental equivalence between linear autoencoders and PCA is a cornerstone in understanding the representational capabilities of these models. Early theoretical work established that a single-layer linear AE, trained with mean squared error (MSE) reconstruction loss, learns a subspace equivalent to that spanned by the principal components of its input data. Bourlard & Kamp

(1988) demonstrated this connection by showing that the linear AE solution could be derived via Singular Value Decomposition (SVD), effectively recovering the principal subspace. Concurrently, Baldi & Hornik (1989) rigorously proved that the optimal weight matrices of a linear AE span the same subspace as the principal components, characterizing the essential points of the associated loss landscape and showing that the global minimum corresponds to the PCA solution.

Recent theoretical analyses have further refined our understanding of linear autoencoders, particularly in terms of optimization dynamics, regularization, and invariance. Work by Kunin et al. (2019) explored the loss landscapes of regularized linear autoencoders, revealing how different regularization schemes affect the geometry and the convergence paths towards the PCA solution. Similarly, Plaut (2018) provided a detailed analysis demonstrating how linear autoencoders, even without explicit orthogonality constraints, recover the principal components themselves (not just the subspace) under specific conditions related to weight initialization and optimization trajectory. The issue of rotational invariance inherent in the basic linear autoencoder objective was addressed by Oftadeh et al. (2020), who proposed a modified loss function to eliminate this invariance and ensure convergence directly to the ordered principal components. The convergence dynamics, especially the role of regularization in guaranteeing eventual recovery of principal components, were further formalized by Bao et al. (2020), solidifying the theoretical link under practical training regimes.

Extending the PCA paradigm beyond linearity has been a major focus, aiming to capture complex, nonlinear structures while retaining desirable properties like ordered, uncorrelated representations. Kernel PCA (Schölkopf et al., 1997) provides a direct nonlinear generalization by implicitly mapping data into a high-dimensional feature space where linear PCA is performed. While powerful, kernel PCA faces scalability challenges with large datasets. Nonlinear autoencoders offer an alternative pathway. Early hierarchical approaches (Scholz & Vigário, 2002) laid the groundwork for nonlinear PCA using multi-layer networks. A significant challenge for standard nonlinear autoencoders is the lack of inherent ordering or orthogonality in their latent dimensions, unlike PCA. To address this, techniques like nested dropout (Rippel et al., 2014) enforce an ordered variance structure during training, compelling the autoencoder to learn features of monotonically decreasing importance, analogous to principal components. More recently, explicit architectural designs have been proposed to bridge deep learning and PCA principles. The PCA-AE framework (Pham et al., 2022) directly incorporates PCA objectives within the autoencoder training process, structuring the latent space to mimic PCA properties while extending them to nonlinear representations through neural networks. Implicit Rank-Minimizing Autoencoder(IRMAE) learns a low-rank representation by only adding several linear layers after the encoder part due to the implicit bias of gradient descent in deep linear networks. Another line of work approaches the "ordering/relevance" issue via probabilistic priors. For instance, ARD-VAE (Automatic Relevance Detection VAE) replaces the fixed prior in VAE with a hierarchical prior over latent dimensions, thereby letting the model automatically infer which latent axes are "active" or relevant.

These efforts collectively highlight the enduring influence of PCA on representation learning, from the well-established theory of linear AEs to ongoing innovations in designing nonlinear autoencoders that preserve the interpretability and ordered structure characteristic of PCA.

## 3 PRELIMINARY

Throughout this work, we always assume the dataset $\mathbf{X}$ to be zero-centered, i.e., each column has zero sample mean.

### 3.1 PRINCIPAL COMPONENT ANALYSIS

Principal Component Analysis (PCA) performs an orthogonal linear transformation on a real inner product space, mapping the data to a new coordinate system. Within this system, the direction of maximum variance in the data aligns with the first coordinate, termed the first principal component, followed by the direction of the next greatest variance on the second coordinate, and so forth. Formally, consider a data matrix $\mathbf{X} \in \mathbb{R}^{p \times n}$ with row-wise zero mean, where each of the $n$ columns represents a different sample and each of the $p$ rows corresponds to a distinct feature. The covariance matrix is defined as:

$$\mathbf{\Sigma} = \mathbf{X}\mathbf{X}^{\top}. \tag{1}$$

PCA can then be defined as a sequential optimization problem or a single optimization problem[1]:

**Sequential optimization**   Principal components are computed by iteratively solving a sequence of optimization problems:
$$\mathbf{u}_k \in \operatorname*{argmin}_{\mathbf{u} \in \mathbb{R}^p, \|\mathbf{u}\|=1} \|\mathbf{X}_{(k)} - \mathbf{u}\mathbf{u}^\top \mathbf{X}_{(k)}\|_F^2, \tag{2}$$
where $\mathbf{X}_{(1)} = \mathbf{X}$ and for $k = 2, \cdots, p$,
$$\mathbf{X}_{(k)} = \mathbf{X}_{(k-1)} - \mathbf{u}_i\mathbf{u}_i^\top \mathbf{X}_{(k-1)}. \tag{3}$$
The vector $\mathbf{u}_i$ represents the $i^{\text{th}}$ principal component, indicating the direction that accounts for the $i^{\text{th}}$ greatest variance in the data. As per the principles of linear algebra, $\mathbf{u}_i$ is the eigenvector associated with the $i^{\text{th}}$ largest eigenvalue $\lambda_i := \mathbf{u}_i^\top \mathbf{\Sigma}\mathbf{u}_i$ of the covariance matrix $\mathbf{\Sigma}$.

**Single optimization**   With a predetermined $d$ $(d \le p)$, one can compute the $d$-dimensional subspace spanned by the first $d$ principal components via
$$\mathbf{U}_d \in \operatorname*{argmin}_{\mathbf{U} \in \mathbb{R}^{p \times d}, \mathbf{U}^\top \mathbf{U} = \mathbf{I}_d} \|\mathbf{X} - \mathbf{U}\mathbf{U}^\top \mathbf{X}\|_F^2. \tag{4}$$
The columns of $\mathbf{U}_d$ constitute an orthonormal basis of the subspace spanned by $\mathbf{u}_1, \cdots, \mathbf{u}_d$.

## 3.2 LINEAR AUTOENCODER AS PCA

An autoencoder is a pair of parametrized neural networks $(\mathcal{E}_\theta, \mathcal{D}_\phi)$, where $\mathcal{E}_\theta : \mathbb{R}^p \to \mathbb{R}^d$ and $\mathcal{D}_\phi : \mathbb{R}^d \to \mathbb{R}^p$ are called encoder and decoder respectively. The optimization objective of an autoencoder is to minimize the reconstruction loss (mean square error):
$$\mathcal{L}_{\text{recon}} = \frac{1}{n} \sum_{i=1}^n \|\mathbf{x}_i - \mathcal{D}_\phi \circ \mathcal{E}_\theta(\mathbf{x}_i)\|_2^2, \tag{5}$$
which can be regarded as a non-linear extension of equation 4. A linear autoencoder is a special case of an autoencoder where both the encoder and decoder are linear transformations, i.e.,
$$\mathcal{E}_\theta(\mathbf{x}) = \mathbf{A}\mathbf{x}, \ \ \forall \mathbf{x} \in \mathbb{R}^p,$$
$$\mathcal{D}_\phi(\mathbf{z}) = \mathbf{B}\mathbf{z}, \ \ \forall \mathbf{z} \in \mathbb{R}^d,$$
for some matrices $\mathbf{A} \in \mathbb{R}^{d \times p}$ and $\mathbf{B} \in \mathbb{R}^{p \times d}$. The autoencoder objective then becomes
$$\mathbf{A}_*, \mathbf{B}_* \in \operatorname*{argmin}_{\mathbf{A} \in \mathbb{R}^{d \times p}, \mathbf{B} \in \mathbb{R}^{p \times d}} \|\mathbf{X} - \mathbf{B}\mathbf{A}\mathbf{X}\|_F^2. \tag{6}$$
It is well known that a linear autoencoder with the above reconstruction cost is closely related to PCA (Bourlard & Kamp, 1988). Indeed, $(\mathbf{A}_*, \mathbf{B}_*)$ constitutes an optimal solution if and only if it is of the form $(\mathbf{A}_*, \mathbf{B}_*) = (\mathbf{Q}\mathbf{U}_d^\top, \mathbf{U}_d\mathbf{Q}^{-1})$ for some invertible matrix $\mathbf{Q} \in \mathbb{R}^{d \times d}$ (Baldi & Hornik, 1989). Moreover, introducing a uniform $\ell_2$ regularization breaks the symmetry of the loss landscape from the full general linear group $\mathrm{GL}_d(\mathbb{R})$ to the orthogonal group $\mathrm{O}_d(\mathbb{R})$. Building on this insight, recent works have shown that one can further constrain the autoencoder to learn the ordered, axis-aligned principal components directly by employing non-uniform $\ell_2$ regularization (Bao et al., 2020) or modified loss functions (Rippel et al., 2014; Oftadeh et al., 2020). Although these schemes exactly recover PCA in the linear regime, they do not extend to non-linear autoencoders. In the non-linear case, the residual variance is entangled with the learned representation. It cannot be attributed to orthogonal directions in input space, so it cannot be systematically accounted for as in PCA.

## 3.3 ISOMETRIC MAPPING

**Definition 1** (Isometry). Let $\mathcal{M}, \mathcal{N}$ be two metric spaces with metric (distance) $d_\mathcal{M}$ and $d_\mathcal{N}$. A mapping $T : \mathcal{M} \to \mathcal{N}$ is called isometric if for any $\mathbf{x}, \mathbf{y} \in \mathcal{M}$,
$$d_\mathcal{M}(\mathbf{x}, \mathbf{y}) = d_\mathcal{N}(T(\mathbf{x}), T(\mathbf{y})). \tag{7}$$

In our work, we assume $\mathcal{M} \subset \mathbb{R}^p$ is a Riemannian manifold and $\mathcal{N} = \mathbb{R}^d$. In this case, $d_\mathcal{M}$ corresponds to the geodesic distance on $\mathcal{M}$ and $d_\mathcal{N}$ corresponds to the Euclidean distance $\|\cdot\|_2$.

---

[1]The objective can be formulated as either maximizing variance or minimizing reconstruction error. We only present the latter since these two formulations are equivalent.

# 4 METHODOLOGY

Although the objective of an autoencoder can be viewed as a nonlinear extension of the PCA optimization problem (equation 4), it lacks the interpretability that PCA offers. In PCA, the projection directions have a clear geometric meaning—they correspond to the directions of maximum variance in the data. In contrast, the representations learned by an autoencoder are often difficult to interpret, as the latent dimensions do not necessarily correspond to meaningful features.

## 4.1 FAILURE OF PCA-AE AND HIERARCHICAL AUTOENCODER

**PCA-AE** PCA-AE (Pham et al., 2022) aims to construct an ordered and disentangled latent space by combining sequential training with covariance regularization. The model first compresses the input into a one-dimensional bottleneck to capture the most significant variation, then progressively expands the latent dimensionality by adding new units while keeping the previously learned ones fixed. To further reduce redundancy, a covariance penalty is applied so that different latent units become as uncorrelated as possible. Despite these design choices, the method faces fundamental difficulties in the nonlinear setting. First, although features may be learned sequentially, the resulting coordinates are neither orthogonal nor strictly uncorrelated, breaking the variance-ordering principle of PCA and allowing information to leak from early to later units. Second, the nonlinear analogues of "principal curves" that the procedure attempts to recover may not exist or may be non-unique for general data distributions, leaving the training process to pursue ill-defined targets.

**Hierarchical Autoencoder** Hierarchical autoencoders (Gorban et al., 2008; Rippel et al., 2014) attempt to impose an ordering on latent coordinates by training the model to reconstruct the input using progressively larger prefixes of the latent vector. Formally, let $\mathcal{E}_\theta^{(k)}$ denote the encoder restricted to the first $k$ coordinates, i.e., for $\mathbf{z} = \mathcal{E}_\theta(\mathbf{x}) = (z_1, \ldots, z_d)$ we define $\mathcal{E}_\theta^{(k)}(\mathbf{x}) = (z_1, \ldots, z_k, 0, \ldots, 0)$. The training objective is a weighted sum of reconstruction errors,

$$\mathcal{L}_{\text{HAE}} = \sum_{k=1}^{d} \alpha_k \, \mathcal{L}_k, \quad \mathcal{L}_k = \sum_{i=1}^{n} \|\mathbf{x}_i - \mathcal{D}_\phi \circ \mathcal{E}_\theta^{(k)}(\mathbf{x}_i)\|_2^2, \tag{8}$$

where $\mathcal{L}_k$ measures the error when only the first $k$ latent components are used. Despite its appeal, HAE faces two fundamental limitations. First, because $\mathcal{L}_k$ monotonically decreases with $k$, the early reconstruction losses $\mathcal{L}_1, \mathcal{L}_2, \ldots$ dominate the objective, causing gradients from later components to be relatively weak. This biases training toward refining the first few latent coordinates while neglecting subsequent ones. Second, since the gradient of $\mathcal{L}_{\text{HAE}}$ is the sum of all partial gradients, the computational cost scales linearly with latent dimensionality $d$, leading to significant inefficiency in high-dimensional settings. These issues hinder both the effectiveness and scalability of HAE.

## 4.2 PRINCIPAL COMPONENT AUTOENCODER

The limitations of PCA-AE and HAE suggest two key lessons: (i) all latent coordinates should be learned jointly rather than sequentially, and (ii) reconstruction from partial intermediate representations introduces inefficiencies and should be avoided. These insights motivate us to design a more direct objective that explicitly enforces the ordering of principal components within the latent space.

We begin with the linear case. Let $\mathbf{Z} = \mathbf{U}^\top \mathbf{X}$ be the representation of data $\mathbf{X}$ under an orthonormal transformation $\mathbf{U}^\top \in \mathbb{R}^{p \times p}$. Denote the variance of the $i^{\text{th}}$ coordinate of $\mathbf{Z}$ by $\sigma_i^2$. The goal is to ensure that $\sigma_1^2$ captures the largest variance, $\sigma_2^2$ the second largest, and so on. To encode this "rank-ordering" preference into a scalar objective, we penalize variance losses more heavily when they occur in later coordinates. Concretely, we assign a strictly increasing sequence of non-negative weights $0 < \gamma_1 < \gamma_2 < \cdots < \gamma_p$ to the coordinates and minimize the weighted sum of variances, $\sum_{i=1}^{p} \gamma_i \sigma_i^2$. This is equivalent to scaling the $i^{\text{th}}$ coordinate of $\mathbf{Z}$ by $\gamma_i^{1/2}$ and summing the variances across all coordinates:

$$\sum_{i=1}^{p} \gamma_i \sigma_i^2 = \text{Tr}\left(\text{Cov}(\mathbf{\Gamma}^{1/2}\mathbf{Z})\right) = \text{Tr}\left(\mathbf{\Gamma}^{1/2}\mathbf{U}^\top \mathbf{\Sigma} \mathbf{U} \mathbf{\Gamma}^{1/2}\right), \tag{9}$$

where $\mathbf{\Gamma} = \mathrm{diag}(\gamma_1, \ldots, \gamma_p)$. By construction, this objective focuses optimization on maximizing early variances: a reduction in $\sigma_1^2$ is weighted least, while the same reduction in a later coordinate incurs a larger penalty. Thus, the solution is implicitly steered toward a descending variance order without requiring explicit enforcement. Although heuristically motivated, the following theorem shows that this objective recovers the principal components in the correct order:

**Theorem 1.** *Let $\mathbf{\Sigma}$ be the covariance matrix of $\mathbf{X}$ with eigenvalues $\lambda_1 \geq \cdots \geq \lambda_p \geq 0$, and let $\mathbf{\Gamma} = \mathrm{diag}(\gamma_1, \ldots, \gamma_p)$ be diagonal with $0 \leq \gamma_1 < \cdots < \gamma_p$. Then the minimum of*

$$\min_{\mathbf{U} \in \mathbb{R}^{p \times p}} \mathrm{Tr}\left(\mathbf{\Gamma}^{1/2}\mathbf{U}^\top \mathbf{\Sigma}\mathbf{U}\mathbf{\Gamma}^{1/2}\right) \quad s.t. \quad \mathbf{U}^\top \mathbf{U} = \mathbf{I} \tag{10}$$

*is $\sum_{i=1}^{p} \lambda_i \gamma_i$. Moreover, $\mathbf{U}_*$ is optimal if and only if its $i^{th}$ column is a unit eigenvector of $\mathbf{\Sigma}$ associated with $\lambda_i$.*

*Proof.* See Appendix B.1. $\qquad\square$

We now extend the above "rank–ordered variance" principle to the nonlinear setting. Suppose the data lie on a Riemannian manifold $\mathcal{M} \subset \mathbb{R}^p$. Let $T : \mathcal{M} \to \mathbb{R}^p$ be an isometric embedding, and define $\mathbf{z} = T(\mathbf{x})$. Because isometries preserve distance, and hence local scales and variances, the coordinates of $\mathbf{z}$ can still be interpreted as orthogonal axes of variation on $\mathcal{M}$. Denoting the variance of the $i^{th}$ coordinate by $\sigma_i^2 = \mathrm{Var}(z_i)$, we again assign strictly increasing non-negative weights $0 < \gamma_1 < \cdots < \gamma_p$ and minimize the weighted total variance $\mathcal{L}_{\mathrm{var}} = \sum_{i=1}^{p} \gamma_i \sigma_i^2$ over all admissible isometric embeddings.

In the autoencoder setting, the mapping $T$ is realized by the encoder $\mathcal{E}_\theta$. We therefore augment the standard reconstruction loss with the weighted-variance penalty above. Crucially, however, an **isometry constraint** is required: without distance preservation, variances in latent space no longer carry geometric meaning. As in prior autoencoder variants, we enforce this constraint softly via regularization rather than a hard constraint:

$$\mathcal{L}_{\mathrm{iso}} = \mathbb{E}\big[\ell(d_{\mathcal{M}}(\mathbf{x}, \mathbf{y}), \|\mathcal{E}_\theta(\mathbf{x}) - \mathcal{E}_\theta(\mathbf{y})\|)\big], \tag{11}$$

where $\ell$ is a loss function. Putting these pieces together, the objective of our *Principal-Component Autoencoder* (PCAE) is

$$\mathcal{L}_{\mathrm{PCAE}} = \mathcal{L}_{\mathrm{recon}} + \beta\big(\mathcal{L}_{\mathrm{var}} + \mathcal{L}_{\mathrm{iso}}\big), \tag{12}$$

with weighting coefficient $\beta > 0$.

**The choice of $\ell$ and $\gamma$.** The variance term tends to contract latent codes toward zero, so careful design of $\ell$ and $\{\gamma_i\}$ is essential for preserving isometry. Below provides principled guidance:

**Theorem 2.** *Let $0 < \gamma_1 < \cdots < \gamma_p < 2$, and let $f^*$ minimize*

$$\mathcal{R}(f) = \mathbb{E}\big[\,\big|\|f(\mathbf{X}) - f(\mathbf{Y})\|^2 - d_{\mathcal{M}}(\mathbf{X}, \mathbf{Y})^2\big|\,\big] + \sum_{i=1}^{p} \gamma_i \mathrm{Var}[f(\mathbf{X})_i]. \tag{13}$$

*Then $\|f(\mathbf{X}) - f(\mathbf{Y})\| = d_{\mathcal{M}}(\mathbf{X}, \mathbf{Y})$ almost surely. Furthermore, if $f^*$ is continuous, the equality holds everywhere.*

*Proof.* See Appendix B.2. $\qquad\square$

Guided by Theorem 2, we set $\ell(a, b) = |a^2 - b^2|$ and choose $\{\gamma_i\}$ satisfying $0 < \gamma_1 < \cdots < \gamma_p < 2$.

### 4.3 DETERMINING THE INTRINSIC DIMENSION

The latent representation learned by our model is inherently ordered: the $i^{th}$ coordinate corresponds to the $i^{th}$ principal component. This structure allows us to estimate the intrinsic dimension in the same way as PCA, using a cumulative variance criterion. Concretely, given a threshold $\tau$ (e.g., 99%), we select the smallest $k$ such that the first $k$ coordinates together explain at least $\tau$ of the total variance. For baseline autoencoders, however, the variance of latent coordinates does not directly correspond to the data variance. In these cases, we adopt the *relevance score* proposed by Saha et al. (2025) to assess the importance of each coordinate and determine the effective latent dimension.

## 5 EXPERIMENTAL RESULTS

**Datasets.** We evaluate our model on both synthetic and real-world datasets to cover scenarios with known and unknown intrinsic structure. The dSprites (Matthey et al., 2017) and 3DShapes (Burgess & Kim, 2018) are synthetic datasets with explicitly controlled generative factors, providing reliable ground truth for evaluating whether a model can recover intrinsic dimensionality. In contrast, MNIST (LeCun et al., 2010) and CelebA (Liu et al., 2015) are real-world image datasets where the underlying generative factors are unknown and must be inferred implicitly. This combination allows us to test both identifiability under controlled settings and robustness in practical, high-variability data. Further dataset details are provided in Appendix C.1.

**Competing Methods.** We benchmark against four representative approaches to ordered/compact latent structure: (i) PCA-AE (Pham et al., 2022), which trains latent units sequentially (from 1D upward) and adds a covariance penalty to reduce redundancy; (ii) Hierarchical Autoencoder (HAE) (Gorban et al., 2008; Rippel et al., 2014), which imposes an order by minimizing a weighted sum of reconstruction losses from latent prefixes; (iii) ARD-VAE (Saha et al., 2025), a Bayesian VAE with automatic relevance determination that scores and prunes latent dimensions; and (iv) IR-MAE (Jing et al., 2020), which encourages low-rank latent codes via implicit rank minimization.

**Implementing Details.** By Theorem 1 and Theorem 2, learning an ordered representation requires enforcing $0 < \gamma_1 < \cdots < \gamma_p < 2$. A naive arithmetic or geometric progression for $\{\gamma_i\}$ is problematic: the former yields nearly identical weights when $p$ is large, while the latter suffers from precision underflow. Both lead to vanishing gradients and slow convergence. To overcome this, we introduce a dynamic-coefficients scheme. We initialize $\gamma_i = 1.9i/p$ and maintain a threshold $t$. Every $K$ epochs ($K = 10$ in our experiments), we identify the smallest index $j$ such that $\sum_{i=1}^{j} \sigma_i^2 > t \cdot \sum_{i=1}^{p} \sigma_i^2$, and update coefficients as

$$\gamma_i = \begin{cases} 0.5i/(j-1), & i < j, \\ 1, & i = j, \\ 1 + 0.5(i-j)/(p-j), & i > j. \end{cases}$$

This adaptive reweighting ensures progressive adjustment of coordinate importance in line with variance allocation, preventing gradient collapse and significantly accelerating training. Model architectures and hyperparameters are listed in Appendix C.2.

### 5.1 DATA WITH KNOWN INTRINSIC DIMENSION

We first validate PCAE on datasets where the ground-truth intrinsic dimension is known. The dSprites dataset has five generative factors (shape, scale, orientation, position-X, position-Y)[2], while 3DShapes has six factors (floor hue, wall hue, object hue, scale, shape, orientation). Because the *shape* factor is categorical, we fix it in both datasets, yielding ground-truth intrinsic dimensions of 5 for dSprites and 4 for 3DShapes. As shown in Table 1, PCAE is the only method that consistently (std = 0) and exactly recovers the true intrinsic dimension. We set the bottleneck size to 16, but importantly, PCAE's variance estimates remain stable under different bottleneck settings (Figure 1), highlighting its robustness in identifying principal coordinates.

Table 1: Intrinsic dimensions estimated by our model and other baselines with $\tau = 99\%$.

| Dataset | Ground Truth | Ours | HAE | PCA-AE | ARD-VAE | IRMAE |
|---------|--------------|------|-----|--------|---------|-------|
| dSprites | 4 | $4.00 \pm 0.00$ | $7.40 \pm 0.49$ | $15.60 \pm 0.49$ | $5.80 \pm 0.40$ | $6.40 \pm 0.80$ |
| 3dShapes | 5 | $5.00 \pm 0.00$ | $6.20 \pm 0.40$ | $14.40 \pm 0.64$ | $6.00 \pm 0.00$ | $5.20 \pm 0.40$ |

### 5.2 DATA WITH UNKNOWN INTRINSIC DIMENSION

**Dimension Estimation.** We next evaluate on real-world datasets (MNIST and CelebA) where the ground-truth intrinsic dimensions are unknown. Unlike synthetic data, the latent factors here vary widely in scale: some (e.g., lighting or pose) exhibit large variance, while others (e.g., subtle

---

[2]An additional color factor is always fixed to white, making the effective dimension 5.

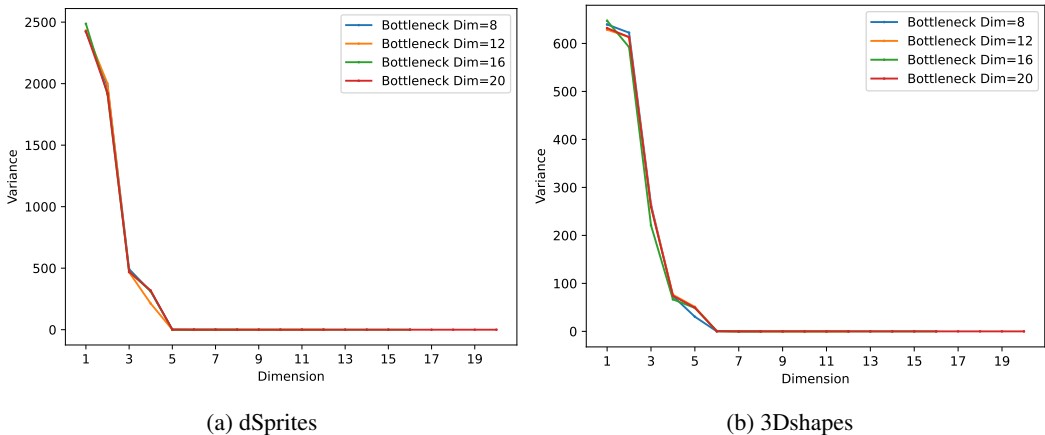

(a) dSprites             (b) 3Dshapes

Figure 1: Estimated variances of latent coordinates for dSprites and 3DShapes under different bottleneck sizes. In both cases, the recovered intrinsic dimension remains fixed at the ground-truth value, demonstrating that PCAE is robust to the choice of bottleneck dimension.

expressions or fine textures) contribute only marginally and may be nearly indistinguishable from noise. To capture this heterogeneity, we estimate intrinsic dimensions under two thresholds $\tau = 99\%, 99.9\%$, which respectively correspond to stricter or looser inclusion of weak factors.

Since no ground truth is available, we report the maximum likelihood estimator (MLE) (Pope et al., 2021), a standard nonlinear intrinsic dimension estimator, as a reference baseline. Specifically, MLE yields values of 11 ($k = 5$) and 13 ($k = 20$) for MNIST, and 17 ($k = 5$) and 26 ($k = 20$) for CelebA.[3] As shown in Table 2, our estimates closely align with these MLE reference ranges, while baseline autoencoders, particularly on CelebA, substantially overestimate the intrinsic dimension. This highlights that PCAE not only avoids overfitting noise but also provides stable and interpretable dimension estimates in complex, high-variability data.

Table 2: Intrinsic dimensions estimated by our model and other baselines for MNIST/CelebA with threshold $\tau = 99\%, 99.9\%$. The bottleneck latent dimension is 24 for MNIST and 64 for CelebA.

| Dataset | $\tau$ | Ours | HAE | PCA-AE | ARD-VAE | IRMAE |
|---------|--------|------|-----|--------|---------|-------|
| MNIST | 99% | 11.00±0.00 | 8.20±0.40 | 23.80±0.40 | 9.20±0.40 | 15.80±0.56 |
| | 99.9% | 14.20±0.40 | 14.00±0.00 | 24.00±0.00 | 11.40±0.49 | 16.80±0.56 |
| CelebA | 99% | 16.00±0.63 | 55.60±1.04 | 61.40±0.49 | 34.80±6.56 | 42.40±0.80 |
| | 99.9% | 27.00±0.89 | 60.60±0.49 | 63.80±0.40 | 54.00±8.00 | 43.00±0.63 |

**Interpolation.** Beyond estimating intrinsic dimension, PCAE also learns interpretable latent spaces that preserve the geometry of the data manifold. A desirable property is *smoothness*: equal steps in latent space should correspond to gradual and consistent changes in the decoded images. Figure 2 illustrates this property—our interpolations vary more uniformly than those of competing methods. To quantify smoothness, we interpolate $l$ steps between two encoded samples, decode each to $\hat{x}^{(t)}$, compute successive distances $d_t = \|\hat{x}^{(t+1)} - \hat{x}^{(t)}\|$, and report $\text{Var}(\{d_t\})$ as the smoothness score (lower is better).[4] As shown in Table 3, PCAE achieves substantially lower variance, indicating smoother and hence more interpretable trajectories through the data manifold.

Table 3: Smoothness (%) evaluated for our model and other baselines. Smaller is better.

| Dataset | Ours | HAE | PCA-AE | ARD-VAE | IRMAE |
|---------|------|-----|--------|---------|-------|
| MNIST | **4.55±0.12** | 33.44±12.61 | 8.00±0.50 | 12.78±1.47 | 5.78±0.29 |
| CelebA | **1.35±0.05** | 1.68±0.07 | 7.85±4.15 | 1.66±0.04 | 1.60±0.04 |

---

[3] $k$ denotes the number of nearest neighbors used by the MLE estimator.

[4] See Appendix for the formal definition.

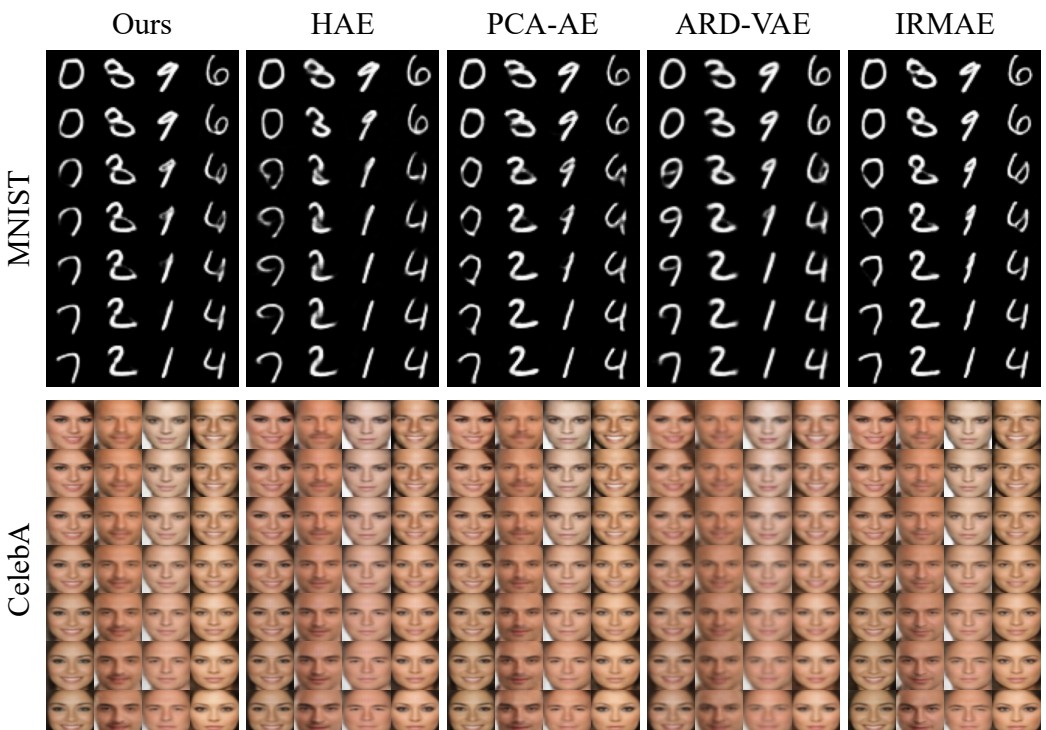

Figure 2: Linear interpolation between two randomly generated samples.

Smoothness alone does not guarantee plausibility—interpolations might drift away from the true distribution. We therefore compute FID (Heusel et al., 2017) between interpolated reconstructions and real samples. Moreover, since PCAE learns a variance-ordered representation, we evaluate FID using only the top-$k$ latent coordinates. Table 4 shows that PCAE captures more meaningful variation with fewer components, yielding the strongest gains when $k$ is small. This demonstrates that ordered latent dimensions directly translate into more faithful and interpretable generative behavior.

Table 4: FID score evaluated for our model and other baselines. Smaller is better.

| Dataset | Dim | Ours | HAE | PCA-AE | ARD-VAE | IRMAE |
|---|---|---|---|---|---|---|
| MNIST | 8 | **56.84±1.62** | 62.44±2.09 | 70.82±3.01 | 64.28±2.03 | 57.97±1.17 |
| | 12 | **49.45±1.28** | 54.35±2.28 | 58.31±2.33 | 57.34±1.95 | 50.55±1.04 |
| | 24 | **45.35±1.37** | 50.49±0.98 | 48.27±1.56 | 48.57±2.11 | 45.97±1.33 |
| CelebA | 16 | **59.42±1.31** | 68.59±1.37 | 82.43±3.25 | 73.55±1.60 | 62.83±1.85 |
| | 32 | **56.99±1.58** | 62.90±1.25 | 73.27±2.77 | 65.60±1.79 | 57.76±0.47 |
| | 64 | 54.82±0.68 | 60.37±0.34 | 66.18±2.38 | 60.42±1.84 | **53.95±0.89** |

## 5.3 Time Complexity

PCA-AE optimizes latent units sequentially, so its training time scales linearly with bottleneck size $p$, i.e. $\mathcal{O}(p)$. HAE also introduces $d$ prefix reconstruction losses, making both its time and space complexity $\mathcal{O}(p)$ (though with a smaller constant factor). In contrast, PCAE, IRMAE, and ARD-VAE optimize all coordinates jointly, so their per-epoch cost is essentially independent of $p$. As shown in Table 5, PCAE is more than an order of magnitude faster than PCA-AE and HAE on CelebA, underscoring its scalability advantage. A one-time cost of PCAE is constructing the geodesic distance matrix over the dataset $\mathbf{X}$, which requires $\mathcal{O}(|\mathbf{X}|^2 \log |\mathbf{X}|)$ time. For large datasets, we approximate by building the graph only on a subset $\tilde{\mathbf{X}} \subset \mathbf{X}$ and linking each $\mathbf{x} \in \mathbf{X}$ to its nearest neighbor $\tilde{\mathbf{x}} \in \tilde{\mathbf{X}}$. Then, whenever one needs the distance between any two points $\mathbf{x}, \mathbf{y} \in \mathbf{X}$, one can approximate it by using $\tilde{\mathbf{x}}, \tilde{\mathbf{y}}$. This reduces the complexity to $\mathcal{O}(|\tilde{\mathbf{X}}|^2 \log |\tilde{\mathbf{X}}| + |\tilde{\mathbf{X}}| \cdot |\mathbf{X}|)$.

Once training is complete, estimating the intrinsic dimension with PCAE requires only computing coordinate-wise variances of latent codes $z$, which takes a few seconds regardless of network size. By contrast, ARD-VAE relies on relevance scores that involve Jacobian evaluation of the decoder with respect to $z$, whose cost scales poorly as model complexity increases. The runtimes reported in Table 5 verify that PCAE achieves both fast training and efficient post-hoc dimension estimation.

Table 5: The training time per epoch of our model and other baselines.

|  | Ours | HAE | PCA-AE | ARD-VAE | IRMAE |
|---|---|---|---|---|---|
| dim = 8 (s/epoch) | 9.09±0.04 | 30.70±0.07 | 58.83±0.03 | 8.88±0.05 | 8.66±0.04 |
| dim = 64 (s/epoch) | 9.19±0.04 | 203.57±0.05 | 464.68±0.02 | 8.99±0.05 | 8.75±0.05 |
| Total Runtime (hrs) | 1.37±0.03 | 14.95±0.02 | 33.10±0.08 | 1.45±0.05 | 1.43±0.05 |

### 5.4 DOWNSTREAM CLASSIFICATION

An important test of learned representations is their utility in downstream tasks. Because PCAE produces an ordered and compact latent space, it is expected to preserve discriminative structure while reducing dimensionality and computational cost. To evaluate this, we freeze the encoder and train a lightweight multilayer perceptron (MLP) classifier on MNIST. The MLP consists of two fully connected layers with 128 hidden units and ReLU activations, optimized with Adam (learning rate $1 \times 10^{-3}$, weight decay $1 \times 10^{-5}$) for 100 epochs. Table 6 shows that PCAE achieves the lowest error rate among all baselines. This confirms that variance-ordered latent coordinates not only provide interpretability but also transfer effectively to practical classification tasks.

Table 6: Downstream classification on MNIST dataset.

| Method | Ours | HAE | PCA-AE | ARD-VAE | IRMAE |
|---|---|---|---|---|---|
| Error(%) | **1.44±0.09** | 1.57±0.14 | 3.76±0.32 | 2.08±0.12 | 1.98±0.11 |

### 5.5 ABLATION STUDY

We finally examine the contributions of the two key loss components in PCAE. Without the isometric term $\mathcal{L}_{\text{iso}}$, distances in latent space no longer reflect the data manifold, making coordinate variances uninterpretable. Without the variance-ordering term $\mathcal{L}_{\text{var}}$, latent dimensions lose their ranking and the intrinsic dimension cannot be identified. Table 7 confirms that both terms are indispensable: only the full objective $\mathcal{L}_{\text{PCAE}}$ recovers the correct intrinsic dimension, whereas removing either constraint leads to severe overestimation or misalignment.

Table 7: Ablation.

| Dataset | Ours | $\mathcal{L}_{\text{var}}$-only | $\mathcal{L}_{\text{iso}}$-only |
|---|---|---|---|
| dSprites | 4.00±0.00 | 12.80±0.40 | 9.80±0.40 |
| 3Dshapes | 5.00±0.00 | 14.40±0.49 | 10.00±0.00 |

## 6 CONCLUSION

We introduced PCAE, a novel autoencoder that extends PCA principles to the nonlinear setting. PCAE learns variance-ordered latent coordinates, enabling post-hoc intrinsic dimension estimation without prior knowledge of the bottleneck size. Experiments demonstrate that PCAE precisely recovers the ground-truth dimensionality on synthetic data, aligns with MLE estimates on real-world datasets, and generates interpretable latent spaces that facilitate smoother interpolations and stronger downstream classification. In addition, PCAE scales efficiently compared to sequential or hierarchical baselines, making it a practical tool for nonlinear representation learning with interpretability guarantees.

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

## A  USAGE OF LARGE LANGUAGE MODELS (LLMS)

We used large language models as a general-purpose writing assistant. Its role was limited to grammar checking, minor stylistic polishing, and improving the clarity of phrasing in some parts of the manuscript. The authors made all substantive contributions to the research and writing.

## B  THEORETICAL PROOF

### B.1  PROOF OF THEOREM 1

Before we formally prove Theorem 1, we first introduce two important lemmas. The first lemma is known as Von Neumann's trace inequality:

**Lemma 1** (Von Neumann). Let $\mathbf{A}, \mathbf{B} \in \mathbb{R}^{p \times p}$ be two square matrices with singular values $\sigma_1(\mathbf{A}) \geq \cdots \geq \sigma_p(\mathbf{A})$ and $\sigma_1(\mathbf{B}) \geq \cdots \geq \sigma_p(\mathbf{B})$ respectively, then

$$\mathrm{Tr}(\mathbf{AB}) \leq \sum_{i=1}^{p} \sigma_i(\mathbf{A})\sigma_i(\mathbf{B}), \tag{14}$$

with equality if and only if $\mathbf{A}$ and $\mathbf{B}^\top$ share singular vectors. Here share singular vectors means that there exists unit vectors $\mathbf{v}_1, \cdots, \mathbf{v}_p, \mathbf{u}_1, \cdots, \mathbf{u}_p \in \mathbb{R}^p$ such that

$$\mathbf{A}\mathbf{v}_i = \sigma_i(\mathbf{A})\mathbf{u}_i, \qquad \mathbf{A}^\top \mathbf{u}_i = \sigma_i(\mathbf{A})\mathbf{v}_i,$$
$$\mathbf{B}^\top \mathbf{v}_i = \sigma_i(\mathbf{B})\mathbf{u}_i, \qquad \mathbf{B}\mathbf{u}_i = \sigma_i(\mathbf{B})\mathbf{v}_i,$$

namely, each $\mathbf{u}_i$ ($\mathbf{v}_i$) is a left (right) singular vector of both $\mathbf{A}$ and $\mathbf{B}^\top$ associated with $\sigma_i(\mathbf{A})$ and $\sigma_i(\mathbf{B})$ respectively.

*Proof.* See (Carlsson, 2021) ∎

The second lemma is a corollary to Lemma 1:

**Lemma 2.** Let $\mathbf{A}, \mathbf{B} \in \mathbb{R}^{p \times p}$ be two positive semi-definite matrices with eigenvalues $\lambda_1(\mathbf{A}) \geq \cdots \geq \lambda_p(\mathbf{A}) \geq 0$ and $\lambda_1(\mathbf{B}) \geq \cdots \geq \lambda_p(\mathbf{B}) \geq 0$ respectively, then

$$\mathrm{Tr}(\mathbf{AB}) \geq \sum_{i=1}^{p} \lambda_i(\mathbf{A})\lambda_{p-i+1}(\mathbf{B}), \tag{15}$$

with equality if and only if the eigenvectors of $\mathbf{A}$ and $\mathbf{B}$ align in reverse order, i.e., there exists unit vectors $\mathbf{u}_i, \cdots, \mathbf{u}_p$ such that each $\mathbf{u}_i$ is a eigenvector of both $\mathbf{A}$ and $\mathbf{B}$ associated with $\lambda_i(\mathbf{A})$ and $\lambda_{p-i+1}(\mathbf{B})$ respectively.

*Proof.* We consider $\alpha\mathbf{I} - \mathbf{B}$ with eigenvalues $\lambda_1(\alpha\mathbf{I} - \mathbf{B}) \geq \cdots \geq \lambda_p(\alpha\mathbf{I} - \mathbf{B})$. It is easy to see that $\lambda_i(\alpha\mathbf{I} - \mathbf{B}) = \alpha - \lambda_{p-i+1}(\mathbf{B})$ and all eigenvectors of $\alpha\mathbf{I} - \mathbf{B}$ associated with $\lambda_i(\alpha\mathbf{I} - \mathbf{B})$ are eigenvectors of $\mathbf{B}$ associated with $\lambda_{p-i+1}(\mathbf{B})$. Taking $\alpha$ large enough such that $\alpha\mathbf{I} - \mathbf{B}$ is still positive semi-definite, i.e. $\alpha - \lambda_p(\mathbf{B}) \geq \cdots \geq \alpha - \lambda_1(\mathbf{B}) \geq 0$, because the singular values are

exactly the eigenvalues of a positive semi-definite matrix, by Lemma 1, we have

$$\text{Tr}(\mathbf{A}(\alpha\mathbf{I} - \mathbf{B})) \leq \sum_{i=1}^{p} \lambda_i(\mathbf{A})\lambda_i(\alpha\mathbf{I} - \mathbf{B})$$

$$= \sum_{i=1}^{p} \lambda_i(\mathbf{A})(\alpha - \lambda_{p-i+1}(\mathbf{B}))$$

$$= \sum_{i=1}^{p} \alpha\lambda_i(\mathbf{A}) - \lambda_i(\mathbf{A})\lambda_{p-i+1}(\mathbf{B}), \tag{16}$$

with equality if and only if $\mathbf{A}$ and $\alpha\mathbf{I} - \mathbf{B}$ share eigenvectors. On the other hand,

$$\text{Tr}(\mathbf{A}(\alpha\mathbf{I} - \mathbf{B})) = \text{Tr}(\alpha\mathbf{A} - \mathbf{A}\mathbf{B})$$

$$= \text{Tr}(\alpha\mathbf{A}) - \text{Tr}(\mathbf{A}\mathbf{B})$$

$$= \sum_{i=1}^{p} \alpha\lambda_i(\mathbf{A}) - \text{Tr}(\mathbf{A}\mathbf{B}), \tag{17}$$

Combing Eq. equation 16 and Eq. equation 17 gives

$$\text{Tr}(\mathbf{A}\mathbf{B}) \geq \sum_{i=1}^{p} \lambda_i(\mathbf{A})\lambda_{p-i+1}(\mathbf{B}) \tag{18}$$

with equality if and only if $\mathbf{A}$ and $\alpha\mathbf{I} - \mathbf{B}$ share eigenvectors, or equivalently, the eigenvectors of $\mathbf{A}$ and $\mathbf{B}$ align in reverse order. $\qquad\square$

Now we can give the proof of Theorem 1.

*Proof.* The objective function can be rewritten using the cyclic property of the trace:

$$\text{Tr}(\mathbf{\Gamma}^{\frac{1}{2}}\mathbf{U}^{\top}\mathbf{\Sigma}\mathbf{U}\mathbf{\Gamma}^{\frac{1}{2}}) = \text{Tr}(\mathbf{U}^{\top}\mathbf{\Sigma}\mathbf{U}\mathbf{\Gamma}). \tag{19}$$

Because $\mathbf{U}$ is orthonormal, $\mathbf{U}^{\top}\mathbf{\Sigma}\mathbf{U}$ has the same eigenvalues as $\mathbf{\Sigma}$, i.e., $\lambda_i(\mathbf{U}^{\top}\mathbf{\Sigma}\mathbf{U}) = \lambda_i(\mathbf{\Sigma}) = \lambda_i$. Meanwhile, we know that $\lambda_i(\mathbf{\Gamma}) = \gamma_{p-i+1}$. Therefore, by Lemma 2,

$$\text{Tr}(\mathbf{U}^{\top}\mathbf{\Sigma}\mathbf{U}\mathbf{\Gamma}) \geq \sum_{i=1}^{p} \lambda_i(\mathbf{U}^{\top}\mathbf{\Sigma}\mathbf{U})\lambda_{p-i+1}(\mathbf{\Gamma}) = \sum_{i=1}^{p} \lambda_i\gamma_i. \tag{20}$$

with equality if and only if the eigenvectors of $\mathbf{U}^{\top}\mathbf{\Sigma}\mathbf{U}$ and $\mathbf{\Gamma}$ align in reverse order.

**Sufficiency** Let $\mathbf{U}_*$ be an orthonormal matrix whose $i^{\text{th}}$ column is an unit eigenvector of $\mathbf{\Sigma}$ associate with $\lambda_i$, then by eigen-decomposition, we have

$$\mathbf{\Sigma} = \mathbf{U}_* diag(\lambda_1, \cdots, \lambda_p)\mathbf{U}_*^{\top}, \tag{21}$$

thus

$$\mathbf{U}_*^{\top}\mathbf{\Sigma}\mathbf{U}_* = \mathbf{U}_*^{\top}\mathbf{U}_* diag(\lambda_1, \cdots, \lambda_p)\mathbf{U}_*^{\top}\mathbf{U}_* = diag(\lambda_1, \cdots, \lambda_p), \tag{22}$$

It is easy to see that $\text{Tr}(diag(\lambda_1, \cdots, \lambda_p)\mathbf{\Gamma}) = \sum_{i=1}^{p} \lambda_i\gamma_i$ and the eigenvector of $diag(\lambda_1, \cdots, \lambda_p)$, $\mathbf{\Gamma}$ align in reverse order.

**Necessity** Now, suppose $\mathbf{U}_*$ is an optimal solution to equation 19, there must exist unit vectors $\mathbf{v}_1, \cdots, \mathbf{v}_p$ such that for $i = 1, \cdots, p$,

$$\mathbf{U}_*^{\top}\mathbf{\Sigma}\mathbf{U}_*\mathbf{v}_i = \lambda_i\mathbf{v}_i, \tag{23}$$

$$\mathbf{\Gamma}\mathbf{v}_i = \gamma_i\mathbf{v}_i. \tag{24}$$

For $i = 1, \cdots, p$, because $\gamma_i$ is distinct from all other $\gamma_j$ ($j \neq i$), Eq. equation 24 implies that $\mathbf{v}_i = \pm\mathbf{e}_i$, where $\mathbf{e}_i$ denotes the $i^{\text{th}}$ standard basis vector. Substituting in Eq. equation 23 yields

$$\mathbf{U}_*^{\top}\mathbf{\Sigma}\mathbf{U}_*\mathbf{e}_i = \lambda_i\mathbf{e}_i, \tag{25}$$

multiplying both sides from the left by $\mathbf{U}_*$, we get

$$\mathbf{\Sigma}\mathbf{U}_*\mathbf{e}_i = \lambda_i\mathbf{U}_*\mathbf{e}_i, \tag{26}$$

hence $\mathbf{U}_*\mathbf{e}_i$ is an unit eigenvector of $\mathbf{\Sigma}$ associate with $\lambda_i$. Since $\mathbf{U}_*\mathbf{e}_i$ is exactly the $i^{\text{th}}$ column of $\mathbf{U}_*$, the proof is completed. $\qquad\square$

### B.2 Proof of Theorem 2

*Proof.* Because $\mathbf{X}, \mathbf{Y}$ are i.i.d., for any scalar function $g$, we have

$$
\begin{aligned}
\mathbb{E}[(g(\mathbf{X}) - g(\mathbf{Y}))^2] &= \mathbb{E}[g^2(\mathbf{X})] + \mathbb{E}[g^2(\mathbf{Y})] - 2\mathbb{E}[g(\mathbf{X})]\mathbb{E}[g(\mathbf{Y})] \\
&= 2\mathbb{E}[g^2(\mathbf{X})] - 2\mathbb{E}[g(\mathbf{X})]^2 \\
&= 2\mathbf{Var}[g(\mathbf{X})].
\end{aligned}
\tag{27}
$$

Applying this to each coordinate of $f$,

$$
\sum_{i=1}^{p} \gamma_i \mathbf{Var}[f(X)_i] = \frac{1}{2}\mathbb{E}[\sum_{i=1}^{p} \gamma_i (f_i(\mathbf{X}) - f_i(\mathbf{Y}))^2] = \frac{1}{2}\mathbb{E}[(f(\mathbf{X}) - f(\mathbf{Y}))^\top \mathbf{\Gamma}(f(\mathbf{X}) - f(\mathbf{Y}))].
\tag{28}
$$

Obviously, $\|f(\mathbf{X}) - f(\mathbf{Y})\| = 0 = d(\mathbf{X}, \mathbf{Y})$ holds whenever $\mathbf{X} = \mathbf{Y}$. Now consider $\mathbf{X} \neq \mathbf{Y}$, let $u := d(\mathbf{X}, \mathbf{Y}) > 0$, $w := f(\mathbf{X}) - f(\mathbf{Y})$ and $v := \|w\|^2$, then

$$
\mathcal{R}(f) = \mathbb{E}[|u - v| + \frac{1}{2}w^\top \mathbf{\Gamma} w].
\tag{29}
$$

Denote $\theta := \frac{w}{\|w\|}$ and $c(\theta) := \frac{1}{2}\theta^\top \mathbf{\Gamma}\theta$, note that $\gamma_1/2 \leq c(\theta) \leq \gamma_p/2$, hence $c(\theta) \in (0, 1)$. Therefore,

$$
|u - v| \geq c(\theta)|u - v| \geq c(\theta)(u - v)
\tag{30}
$$

which implies $|u - v| + c(\theta)v \geq c(\theta)u$ with equality holds if and only if $u = v$. Therefore,

$$
\mathcal{R}(f) = \mathbb{E}[|u - v| + c(\theta)v] \geq \mathbb{E}[c(\theta)u].
\tag{31}
$$

with equality holds if and only if $u = v$, i.e. $\|f(\mathbf{X}) - f(\mathbf{Y})\| = d(\mathbf{X}, \mathbf{Y})$ *a.s.*. Furthermore, we know $d$ is continuous, if $f$ is continuous, the function $h(x, y) := \|f(x) - f(y)\|$ is also continuous. To complete the proof, we only need the fact that two continuous functions that agree on a dense set must agree everywhere. $\square$

## C ADDENDUM TO EXPERIMENTAL RESULTS

### C.1 DATA DESCRIPTION

To evaluate our method, we consider both synthetic datasets with known generative factors and real-world datasets with unknown latent structures. This choice allows us to (i) verify whether our model can correctly estimate intrinsic dimension when ground truth is available, and (ii) test robustness in realistic scenarios where intrinsic factors are complex or unobservable. In addition, the synthetic datasets provide controlled environments for disentanglement analysis, while the real-world datasets enable evaluation of interpolation quality, smoothness, and distributional fidelity.

**dSprites (Matthey et al., 2017).** dSprites consists of 2D binary images ($64 \times 64$) generated from independent latent factors: shape, scale, orientation, $x$-position, and $y$-position. A color factor is constant. To remove categorical ambiguity, we fix the shape factor to be $ellipse$, yielding a ground-truth intrinsic dimension of 4. We also constrained the orientation factor in the range $[0, \frac{28}{39}\pi]$ to avoid a circular structure since it may introduce an extra dimension. The total number of samples available is 92,160, we use 70% as our training set, 15% as validation set and the rest 15% as test set. We use dSprites to test whether our model can exactly recover known latent dimensions and produce ordered representations consistent across different bottleneck sizes.

**3DShapes (Burgess & Kim, 2018).** 3DShapes contains 480,000 RGB images ($64 \times 64$) procedurally rendered with six latent factors: floor hue, wall hue, object hue, scale, shape, and orientation. As the shape factor is categorical, we fix it to be $square$, resulting in an effective intrinsic dimension of 5. Because the variance of floor/wall/object hue is much higher than that of scale/orientation, the proportion of total variance contributed by scale/orientation is far below 1%. To make these two factors distinguishable from noise, we halve the range of the hue components, resulting in a total of 15,000 samples. We use 70% as our training set, 15% as a validation set, and the remaining 15% as a test set.

**MNIST (LeCun et al., 2010).** MNIST is a dataset of 70,000 grayscale handwritten digit images ($28{\times}28$). Unlike synthetic datasets, the underlying factors (e.g., stroke width, style, slant) are not explicitly defined and vary widely in scale, with some resembling noise. This makes MNIST a representative benchmark for real-world data where the ground-truth intrinsic dimension is unknown. We use 60,000 samples as a training set, 5,000 as a validation set, and the remaining 5,000 as a test set. In our experiments, we compare our estimated intrinsic dimension with the maximum likelihood estimator (MLE) baseline and further evaluate latent interpolation smoothness and FID scores to assess the quality of the manifold. MNIST is also used for downstream classification on learned representations.

**CelebA (Liu et al., 2015).** CelebA contains over 200,000 celebrity face images, aligned and cropped to $64{\times}64$ resolution. It exhibits significant variability in attributes such as pose, hairstyle, expression, and illumination. The generative factors are complex and unobservable, making the estimation of intrinsic dimension highly challenging. We use CelebA to assess whether our model avoids overestimating the intrinsic dimension compared to baseline autoencoders. We use 60,000 samples as a training set, 5,000 as a validation set, and 5,000 as a test set. In addition, CelebA provides a realistic setting for evaluating interpolation smoothness and FID, reflecting the semantic fidelity of learned latent spaces under high-dimensional natural image variation.

## C.2 EXPERIMENTAL SETTINGS

### C.2.1 MODEL ARCHITECTURE

The architecture of the encoder and decoder used in each experiment is given in Table 8. $\text{Conv}_n$ / $\text{ConvT}_n$ denote convolutional / transposed-convolutional layers whose output channel dimension is $n$. All convolutional layers use a $4 \times 4$ kernel with stride two and padding 1, and $\text{FC}_n$ denotes a fully connected layer with output dimension $n$.

Table 8: The architecture of the encoder and the decoder for each dataset.

| Dataset | Dsprites | 3DShapes | MNIST | CelebA |
|---|---|---|---|---|
| **Encoder** | $x \in \mathbb{R}^{64 \times 64 \times 1}$ $\to \text{Conv}_{32} \to \text{ReLU}$ $\to \text{Conv}_{32} \to \text{ReLU}$ $\to \text{Conv}_{64} \to \text{ReLU}$ $\to \text{Conv}_{64} \to \text{ReLU}$ $\to \text{flatten} \to \text{FC} \to z \in \mathbb{R}^d$ | $x \in \mathbb{R}^{64 \times 64 \times 3}$ $\to \text{Conv}_{32} \to \text{ReLU}$ $\to \text{Conv}_{32} \to \text{ReLU}$ $\to \text{Conv}_{64} \to \text{ReLU}$ $\to \text{Conv}_{64} \to \text{ReLU}$ $\to \text{flatten} \to \text{FC} \to z \in \mathbb{R}^d$ | $x \in \mathbb{R}^{32 \times 32 \times 1}$ $\to \text{Conv}_{64}+\text{BN} \to \text{ReLU}$ $\to \text{Conv}_{128}+\text{BN} \to \text{ReLU}$ $\to \text{Conv}_{256}+\text{BN} \to \text{ReLU}$ $\to \text{Conv}_{512}+\text{BN} \to \text{ReLU}$ $\to \text{flatten} \to \text{FC} \to z \in \mathbb{R}^d$ | $x \in \mathbb{R}^{64 \times 64 \times 3}$ $\to \text{Conv}_{128}+\text{BN} \to \text{ReLU}$ $\to \text{Conv}_{256}+\text{BN} \to \text{ReLU}$ $\to \text{Conv}_{512}+\text{BN} \to \text{ReLU}$ $\to \text{Conv}_{1024}+\text{BN} \to \text{ReLU}$ $\to \text{flatten} \to \text{FC} \to z \in \mathbb{R}^d$ |
| **Decoder** | $z \in \mathbb{R}^d \to \text{FC}$ $\to \text{reshape } 4 \times 4 \times 64$ $\to \text{ConvT}_{64} \to \text{ReLU}$ $\to \text{ConvT}_{32} \to \text{ReLU}$ $\to \text{ConvT}_{32} \to \text{ReLU}$ $\to \text{ConvT}_1 \text{ (logits)}$ | $z \in \mathbb{R}^d \to \text{FC}$ $\to \text{reshape } 4 \times 4 \times 64$ $\to \text{ConvT}_{64} \to \text{ReLU}$ $\to \text{ConvT}_{32} \to \text{ReLU}$ $\to \text{ConvT}_{32} \to \text{ReLU}$ $\to \text{ConvT}_3 \to \text{Sigmoid}$ | $z \in \mathbb{R}^d \to \text{FC}$ $\to \text{reshape } 2 \times 2 \times 512$ $\to \text{ConvT}_{256}+\text{BN} \to \text{ReLU}$ $\to \text{ConvT}_{128}+\text{BN} \to \text{ReLU}$ $\to \text{ConvT}_{64}+\text{BN} \to \text{ReLU}$ $\to \text{ConvT}_1 \to \text{Sigmoid}$ | $z \in \mathbb{R}^d \to \text{FC (65536)}$ $\to \text{reshape } 8 \times 8 \times 1024$ $\to \text{ConvT}_{512}+\text{BN} \to \text{ReLU}$ $\to \text{ConvT}_{256}+\text{BN} \to \text{ReLU}$ $\to \text{ConvT}_{128}+\text{BN} \to \text{ReLU}$ $\to \text{ConvT}_3 \to \text{Sigmoid}$ |

### C.2.2 HYPERPARAMETERS

The training hyperparameters used for each experiment are listed in Table 9.

The hyperparameters for PCAE, IRMAE, and ARD-VAE are selected by grid search or original work and listed in Table 10. HAE and PCAAE have no hyperparameters.

Table 9: Training hyperparameters

| Hyperparameter | dSprites | 3Dshapes | MNIST | CelebA |
|---|---|---|---|---|
| learning rate | 1e-4 | 1e-4 | 1e-4 | 1e-4 |
| epochs | 1000 | 1000 | 500 | 250 |
| batch size | 256 | 256 | 256 | 256 |

Table 10: Hyperparameters for PCAE

| Model | Parameter | dSprites | 3Dshapes | MNIST | CelebA |
|---|---|---|---|---|---|
| PCAE | $\beta$ | 5e-3 | 5e-2 | 1e-4 | 1e-4 |
| IRMAE | $l$ | 4 | 4 | 4 | 4 |
| ARD-VAE | $\beta$ | 5.0 | 5.0 | 0.5 | 1.0 |

## C.3 COMPUTATION OF SMOOTHNESS

In the main paper, we use $smoothness$ to evaluate the quality of the learned latent representation. It is defined (calculated) as follows:

1. Randomly sample $N$ pairs of test images.
2. For each pair $(x_i, x_j)$, encode them into the latent space: $(z_i, z_j)$.
3. Along the straight line in latent space, produce $m$ equally spaced intermediate codes $z_i = z^{(0)}, \cdots, z^{(m)} = z_j$, where $z^{(t)} = (1 - \frac{t}{m})z_i + \frac{t}{m}z_j$.
4. Decode each $z^{(t)}$ back to image $\hat{x}^{(t)}$.
5. Compute the inter-step distances: $d_t = \|\hat{x}^{(t)} - \hat{x}^{(t-1)}\|$, $t = 1, \cdots, m$.
6. Compute variance of $d_1, \cdots, d_m$, denotes as $var(x_i, x_j)$.
7. The $Smoothness$ is then defined as the average of $var(x_i, x_j)$ over all $N$ pairs.

## C.4 COMPARISON OF DIFFERENT ISOMETRIC CONSTRAINTS

In the main paper, we demonstrate that $l_{\text{iso}}(d, \hat{d}) = |d^2 - \hat{d}^2|$ is suitable for our PCAE, here we further verify its distinctiveness by comparing it with other loss function: $1. l_{\text{square}}(d, \hat{d}) = (d - \hat{d})^2$; $2. l_{\text{log}}(d, \hat{d}) = (\log(\frac{d}{\hat{d}}))^2$. The results are reported in Table 11.

Table 11: Intrinsic dimension estimated by PCAE with different isometric constraints.

| | dSprites | 3Dshapes | MNIST | | CelebA | |
|---|---|---|---|---|---|---|
| | $\tau$=99% | $\tau$=99% | $\tau$=99% | $\tau$=99.9% | $\tau$=99% | $\tau$=99.9% |
| $\ell_{\text{iso}}$ | 4.00±0.00 | 5.00±0.00 | 11.00±0.00 | 14.20±0.40 | 16.00±0.63 | 27.00±0.89 |
| $\ell_{\text{square}}$ | 5.20±0.40 | 6.40±0.49 | 6.00±0.00 | 11.20±0.40 | 11.20±0.40 | 18.80±0.56 |
| $\ell_{\text{log}}$ | 8.00±0.00 | 3.80±0.40 | 9.60±0.49 | 20.80±0.40 | 44.60±1.04 | 61.00±0.63 |

