# OpenReview forum: "Learning Ordered Representations in Latent Space for Intrinsic Dimension Estimation via Principal Component Autoencoder"
_ICLR.cc/2026/Conference — ICLR 2026 Conference Withdrawn Submission_

### Official Review · Reviewer_KnMm · 2025-10-28

**Soundness:** 1
**Presentation:** 3
**Contribution:** 2
**Rating:** 2
**Confidence:** 5

**Summary:**

The paper aims to derive a formulation for autoencoders to perform nonlinear PCA. The main idea is to enforce the encoder to be isometric and to weight the latent variances to induce ordering. That weighting works theoretically (at least in the linear case) due to Von Neumann’s trace inequality. The authors then extrapolate this linear formulation to nonlinear ones by replacing the orthogonality constraint on the linear encoder with the isometry constraint on the nonlinear encoder.

**Strengths:**

1. The paper presents a novel method supported by rigorous theoretical analysis, offering some valuable insights, especially Lemma 2.
2. The paper is well-written with good structure, easy to follow.

**Weaknesses:**

Overall, the authors should conduct experiments more rigorously and exercise caution when making claims. Their reliance on MLE for dimension estimation is the main flaw in their result. Some components of their method are also well-known easy victims of the curse of dimensionality. These combined undermine the soundness of this work.

# Major
1. Many experimental results are unsound and accompanied by overly strong claims. Some flaws may even undermine the method’s legitimacy.
   1.  PCAE's robustness to the choice of bottleneck dimension. I don't think the toy results in Fig. 1 is convincing. I wonder what would happen if the bottleneck dimension is set at least an order of magnitude larger than the intrinsic dimension, and the dataset's intrinsic dimensionality's influence on that performance. The author should perform the same experiment on MNIST and CelebA, and choose much larger max bottleneck dimension (for instance, 100 for dsprites and 3Dshapes, 500 for MNIST, 2000 for CelebA). It's necessary because it's often hard to guess the true dim of high dimensional real world data, so we need to select higher upper bound.
   2. There are several defects in the investigation of dimension estimation:
      1. The MLE estimator Pope adopted does not work well in very high dimensional setting, because it relies on nearest neighbors, which suffers from the curse of dimensionality. In fact MLE is known to significantly underestimate the true intrinsic dimension when it's large. So we should take its result on high-dim datasets like CelebA with a grain of salt, and the claim in Sec. 5.2 that "other methods substantially overestimate the dimension" might instead indicate a defect in PCAE. See:
         * _Levina, Elizaveta, and Peter Bickel. "Maximum likelihood estimation of intrinsic dimension." Advances in neural information processing systems 17 (2004)._
            * Even the original paper itself mentioned MLE's negative bias.
         * _Ceruti, Claudio, et al. "Danco: An intrinsic dimensionality estimator exploiting angle and norm concentration." Pattern recognition 47.8 (2014): 2569-2581._
         * _Gomtsyan, Marina, et al. "Geometry-aware maximum likelihood estimation of intrinsic dimension." Asian Conference on Machine Learning. PMLR, 2019._
         * _Erba, Vittorio, Marco Gherardi, and Pietro Rotondo. "Intrinsic dimension estimation for locally undersampled data." Scientific reports 9.1 (2019): 17133._
      2. The latent space can be compressed more at the expense of reconstruction quality. It's a trade-off. So whenever investigating the dimension reductions capability of different AEs, their reconstruction error must also be reported. A good approach is to vary the regularization weight in the loss function, train the AEs for different weights to obtain different reconstruction–dimension trade-offs, and plot their resulting points on a 2D reconstruction–dimension graph for comparison. This might be a better option than MLE since it avoids its unreliable estimation. See this work that also aims to derive nonlinear PCA:
         * _Chen, Qiuyi, and Mark Fuge. "Compressing latent space via least volume." International Conference on Learning Representations.  2024._
   3. MNIST alone is not sufficient to confirm PCAE's help in downstream classification. The last statement of Sec. 5.4 is too strong.
   4. For each application, the relevance score of each individual latent dimension should be plotted to prove they are ordered by their importance to the data variation.

2. Some notable limitations of this method:
   1. This method relies on the estimated geodesic distance matrix to enforce the isometry constraint. Though not clearly mentioned in this paper, I assume it's the method used in Isomap or something similar. This neighborhood-graph-based method also suffers from the curse of dimensionality just like MLE, and it will be even more extreme when the NN method at the end of page 8 is used for high dimensional settings. That is why many researchers use Jacobian-based method for AE's isometry regularization. See:
      * _Gropp, Amos, Matan Atzmon, and Yaron Lipman. "Isometric autoencoders." arXiv preprint arXiv:2006.09289 (2020)._
      * _Lee, Yonghyeon, et al. "Regularized autoencoders for isometric representation learning." International Conference on Learning Representations. 2022._
    2. The method's behavior should depend heavily on $\beta$. Its effect should be studied.


# Minor
1. The authors had better mention right after Theorem 2 on page 5 how they evaluate $d_\mathcal{M}$. I was very confused until I reach page 8. In addition, it's not clear which exact estimation method they use.
2. Using the hyperparameters from the original works (line 747) may unfairly disadvantage other AEs in comparisons, since those hyperparameters might not have been finely tuned for them in the original works with grid search, or they may not be chosen to beat some SOTA scores.
3. It should be $u\coloneqq d(X, Y)^2$ on line 662.

**Questions:**

See the weaknesses.

---

### Official Review · Reviewer_jLfJ · 2025-10-30

**Soundness:** 1
**Presentation:** 1
**Contribution:** 2
**Rating:** 2
**Confidence:** 4

**Summary:**

This paper proposes a generalisation of Principal Component Analysis to the non-linear setting, using an autoencoder. The goal of this is to achieve intrinsic dimensionality estimation.

**Strengths:**

I found the main, high-level idea of reweighting variances to find intrinsic dimensionality, interesting.

**Weaknesses:**

I found the detail of the reasoning of the paper difficult to follow, I put some examples below.

The biggest weakness I found was the motivation of section 4.2. As far as I can tell, the goal is to take the variances of the components found by the PCA and re-weight them with scalars to penalise variances in the later components. But these scalars are completely arbitrary, and have no link to the data at hand. You could put anything you like there, and all it will do is change how much variability you think is in each dimension, but nothing to do with the real variability in this dimension. So I did not understand how this works for dimension estimation, what says that you have meaningful weightings ?

You are solving a PCA problem where you reweight the variances of the projected data, and try to minimise the sum of the variances. I had a hard time understanding why you wanted to minimise the sum of the variances in Theorem 1, usually when we formulate the PCA, you try and maximise the sum of the variances. This seemed surprising. Furthermore, indeed if you apply the orthogonal matrix $U$ to some data, with no dimensionality reduction (because $U$ is of size $p \times p$), then the sum of the variances of the data are maintained. Usually, it is formulated in the case where $U$ is of size $p \times d$ (or $d \times p$, depending on whether you use row or column vectors). I think you have this theorem to show that by doing your re-weighting, then the eigenvectors and eigenvalues are the same, but then I do not understand what the point of the section is. If you have exactly the same transformation as the PCA, and all you are doing is giving different weights to eigenvalues, but these weights are arbitrary, I do not see how this helps to estimate intrinsic dimensionality (you are using the same transformation as a PCA, nothing changes).

The results are quite limited, the only real-world dataset being celeb-a, and a very small resolution (64x64). I did not see much visual difference between PCA-AE and this method in the visual results, which makes sense because the proposed method is more or less a PCA on an autoencoder. I am not quite sure why these results were shown: indeed, in Theorem 1, you prove that your method finds the eigenvalues and eigenvectors of $\Sigma$ (that is to say, the covariance matrix of the original data), so your results are naturally pretty much the same as an autoencoder and a PCA (PCA-AE).

**Questions:**

- Theorem 2 : you have a norm (ie $\lVert \rVert$) between two scalars (the squared norm of the difference $f(X)-f(Y)$, and $d_{\mathcal{M}(X,Y)}$), why not just put the square, since you have two scalars? This is odd, and hinders the understanding of the equation (we expect vectors instead of scalars).
- Proof of Theorem 2: you say that equality of equation 31 holds iff $u=v$. But I do not understand why equality should necessarily hold. Of course, if it holds, then by definition of the equation, since the variance is a positive number, then $u$ is necessarily equal to $v$. But what says that equality will be reached ? More generally, in equation 13, it seems like you are saying that the minimum possible value of $R(f)$ is reached if the distances are equal. But this is different than saying "let f minimise ...". What if there is no possible world in which the $f$ is good enough ? Then euqation 13 would still be minimised (ie, the global minimum with this $f$ is reached), but you would never reach the minimum you want. Another way of phrasing it: let us minimise, wrt $f$, $\lVert f(X)-Y \rVert^2_2$, then necessarily $f(X)=Y$. This is not true, it totally depends on $f$. We only know that $\lVert f(X)-Y \rVert^2_2 \geq 0$, but not that 0 is reached.
- page 5, line 256: I think you mean $f^\ast$ here.

---

### Official Review · Reviewer_VSKX · 2025-10-31

**Soundness:** 3
**Presentation:** 3
**Contribution:** 3
**Rating:** 6
**Confidence:** 4

**Summary:**

This work proposes PCAE, which addresses the  nonlinear of PCA under the framework of AE with preserved interpretation superiority in linear cases, i.e., ordered representations and explicit explanations on minimal reconstruction/maximal variances. This works also claims the estimation  for intrinsic dimension estimation under the nonlinear setups. The soundness is supposed with som analytical analyses through the presented theorems and numerical evaluations.

In general, this work is interesting and aims to provide in-depth understandings on the long-standing tool of PCA. Even in the era of learning with massive data and giant architectures, it is still be promising to visit these classical and fundamental tools. Nevertheless, the reviewer would also like to note that there are still quite some aspects worthy of further discussions and revisions indeed towards a publishable piece of work.

**Strengths:**

1. The addressed problem is of significance to the community.
2. Good clarity and compactness in context.
3. The method is simple but gives new insights rather than in  a direct manner.

**Weaknesses:**

1. The contribution and the key novelty should be further clarified.
From the title and most of the presented experiments, it seems that intrinsic dimension estimation is positioned as a main part. However, the proposed algorithm and its mechanisms and technical contributions are mainly enclosed  in Sec 4.2, i.e., PCAE. In sec 4.3, the determination of intrinsic dimensions is introduced in short, which is simply taking the conventional technique of choosing a threshold; even it takes the relevance score to consider the latent coordinates in the nonlinear cases, it is neither  really new nor contributes to the main novelty/contribution  of this work, technically at least. It might be confusing that which is the core contribution and the focus. Could the authors make it clear and revise accordingly?
2. The methods covering PCA, KPCA, and  AE can do many things, beyond the dimension estimations. Although the interpolation and classification tasks are presented, their corresponding experiments are a bit preliminary.
3. The “ordered representations” is emphasized. However, in the experiments or (literature review), the evaluations (explanations) on the significance of such property are neither sufficiently presented and exemplified.
4. As the nested dropout can do similar stuff, could it be possible to incorporated it into related series methods of AEs and be compared? Why and why not.
5. In the experiments of interpolation, it might be interesting to conduct similar experiments for some datasets (e.g., Cars3D, 3DShapes, SmallNORB), where the ranked latent features/coordinates can be further explored, as the ground-truth latent factors and their physical explanations are known,
6. What about the disentanglement of the learned latent space? This is commonly conducted in AE-based methods and representation learning.
7. For (12) and such, the reviewer would suggest to note the parameters in the optimization process.
8. Could [1]  be considered as a compared method, as it optimizes a PCA module in the latent space with similar AE architectures.

[1] Pandey, Arun, et al. "Disentangled representation learning and generation with manifold optimization." Neural Computation 34.10 (2022): 2009-2036.

**Questions:**

See the weakness.

---

### Official Review · Reviewer_XJ1u · 2025-10-31

**Soundness:** 2
**Presentation:** 2
**Contribution:** 1
**Rating:** 2
**Confidence:** 4

**Summary:**

This paper introduces an principal components autoencoder framework with a regularizer that enforces isometry of the encoder in the nonlinear setting. The aim is to learn the latent dimensions in descending order of importance, providing interpretability. Experimental results show that this framework is effective for nonlinear dimensionality reduction tasks.

**Strengths:**

1. The studied problem of finding latent components with geometric interpretation is important to the machine learning community.
2. The experimental results look very convincing.

**Weaknesses:**

1. It is difficult to ascertain the correctness of the experiments since the code is not provided in the supplementary material and the experimental setup is not described in sufficient detail (see questions below).
2. The theoretical results and insights are very limited. Moreover, since the isometry is “enforced” via a soft regularization, it is not necessarily true that the resulting encoder will be an isometry. Further, it is not clear why notions such as correlation are useful in a nonlinear setting, since these are tailored to linear PCA.
3. The implemented adaptive reweighting means that the cost function changes every $K$ epochs. There is no reason to believe that this process will converge and it also means that the condition of Theorem 2 is not satisfied.
4. While Theorem 2 provides a choice of $l$ and $\gamma$ such that the isometry condition is satisfied at a minimizer, this does not mean that the variance is also jointly maximized.
5. The computational cost is infeasible for modern ML applications. The authors alleviate this issue by first doing some clustering. However, this already reduces the intrinsic dimensionality so it begs the question what the main driver of the nonlinear dimensionality reduction is.

Minor issues/suggestions:
1. Line 084: Missing citation for IRMAE
2. Line 096: assumes each column has zero mean while Line 104 assumes each row has zero mean
3. Line 128: parametrized -> parameterized
4. Line 134 (and others): provide equation referencing via (4) and not 4.
5. Line 144: $U$ not defined
6. Line 559: Wrong capitalization von Neumann

**Questions:**

1. Regarding the experimental reproducibility: how is the geodesic distance matrix constructed? What manifold $\mathcal{M}$ is used? How is $\textnormal{Var}(f(z_i))$ calculated (is the mean properly taken into account, since $\mathbb{E}[f(z)] \neq f(\mathbb{E}[z]))$?
2. Can you report whether the learned encoder is actually an isometry?
3. The introduction states that kernel PCA faces scalability challenges. However, using e.g. Nyström approximations or random Fourier features, this can drastically reduce their time complexity (to the same complexity as reported in Section 5.3). Could some experiments comparing against these kernel methods also be performed?
4. Table 5 reports the total runtime in hours. Is this correct?
5. Please provide an algorithm block that details the complete algorithm that is implemented, including all the heuristics such as the adaptive $\gamma$ and the nearest neighbor preprocessing.

---

### Note · Authors · 2026-01-18

I have read and agree with the venue's withdrawal policy on behalf of myself and my co-authors.